# Deep Learning Assisted Diagnosis of Onychomycosis on Whole-Slide Images

**DOI:** 10.3390/jof8090912

**Published:** 2022-08-28

**Authors:** Philipp Jansen, Adelaida Creosteanu, Viktor Matyas, Amrei Dilling, Ana Pina, Andrea Saggini, Tobias Schimming, Jennifer Landsberg, Birte Burgdorf, Sylvia Giaquinta, Hansgeorg Müller, Michael Emberger, Christian Rose, Lutz Schmitz, Cyrill Geraud, Dirk Schadendorf, Jörg Schaller, Maximilian Alber, Frederick Klauschen, Klaus G. Griewank

**Affiliations:** 1Department of Dermatology, University Hospital Essen, Hufelandstraße 55, 45122 Essen, Germany; 2Department of Dermatology, University Hospital Bonn, Venusberg-Campus 1, 53127 Bonn, Germany; 3Aignostics GmbH, 10555 Berlin, Germany; 4Department of Dermatology, Charité Berlin, 10117 Berlin, Germany; 5Center for Dermatopathology, 79106 Freiburg, Germany; 6Department of Dermatology Hornheide, 48157 Münster, Germany; 7Dermatopathology Near Mainz, 55268 Nieder-Olm, Germany; 8Dermatohistology am Stachus, 80331 München, Germany; 9Patholab, 5020 Salzburg, Austria; 10Institute for Dermatohistology, 23562 Lübeck, Germany; 11Institute for Dermatopathology, 53115 Bonn, Germany; 12Department of Dermatology, University Hospital Mannheim, 68167 Mannheim, Germany; 13MVZ Dermatopathology Duisburg Essen GmbH, 45329 Essen, Germany; 14Institute of Pathology, Charité Berlin, 10117 Berlin, Germany; 15Institute of Pathology, Ludwig-Maximilians University Munich, 80337 München, Germany; 16German Cancer Research Center (DKFZ) and German Cancer Consortium (DKTK), Munich Partner Site, 80336 München, Germany; 17BIFOLD—Berlin Institute for the Foundations of Learning and Data, 10587 Berlin, Germany; 18BIH—Berlin Institute of Health, Anna-Louisa-Karsch-Straße 2, 10178 Berlin, Germany

**Keywords:** deep learning, artificial intelligence, U-NET, onychomycosis, dermatology

## Abstract

Background: Onychomycosis numbers among the most common fungal infections in humans affecting finger- or toenails. Histology remains a frequently applied screening technique to diagnose onychomycosis. Screening slides for fungal elements can be time-consuming for pathologists, and sensitivity in cases with low amounts of fungi remains a concern. Convolutional neural networks (CNNs) have revolutionized image classification in recent years. The goal of our project was to evaluate if a U-NET-based segmentation approach as a subcategory of CNNs can be applied to detect fungal elements on digitized histologic sections of human nail specimens and to compare it with the performance of 11 board-certified dermatopathologists. Methods: In total, 664 corresponding H&E- and PAS-stained histologic whole-slide images (WSIs) of human nail plates from four different laboratories were digitized. Histologic structures were manually annotated. A U-NET image segmentation model was trained for binary segmentation on the dataset generated by annotated slides. Results: The U-NET algorithm detected 90.5% of WSIs with fungi, demonstrating a comparable sensitivity with that of the 11 board-certified dermatopathologists (sensitivity of 89.2%). Conclusions: Our results demonstrate that machine-learning-based algorithms applied to real-world clinical cases can produce comparable sensitivities to human pathologists. Our established U-NET may be used as a supportive diagnostic tool to preselect possible slides with fungal elements. Slides where fungal elements are indicated by our U-NET should be reevaluated by the pathologist to confirm or refute the diagnosis of onychomycosis.

## 1. Introduction

Onychomycosis numbers among the most common fungal infections in humans with a prevalence of 10% per year in the general population worldwide. The incidence strongly increases in older patients or patients suffering from vascular diseases or an impaired immune system [1]. Localized yellowish discoloration of the finger- or toenail is one of the most common initial clinical symptoms; however, the presentation varies depending on the strain of fungus. Prolonged infection often leads to a clinically visible thickening and dystrophia of the nails [2]. Surveys on therapeutic approaches show that many physicians initiate treatment solely based on the clinical appearance of the nails [3,4].

According to different international guidelines, confirmation of diagnosis is still mandatory before starting systemic antimycotic therapy for refractory or severe clinical courses. Reliable diagnosis of fungus is necessary to exclude inflammatory, hereditary, or metabolic disorders and neoplasms [5,6], which are difficult to exclude solely based on clinical examination. Currently, there is no acknowledged standard diagnostic assay, but five methods are widely used: direct microscopy using potassium hydroxide (KOH) staining, fluorescence optical preparation, culture assays, PCR, and histologic analysis using PAS stains. Direct microscopy including KOH preparation is a low-cost and rapid diagnostic approach but may suffer from low sensitivity compared with culture and histology [7,8]. Fluorescence whiteners (Blancophor or Calcofluor preparation) bind to chitin and cellulose, and fluoresce when exposed to UV light. This approach shows higher sensitivity compared with KOH preparation but is still dependent on the experience of the examiner [9]. Culture assays provide information on fungal species but can take 4 weeks or longer to obtain results. Cultural growth requires a minimum number of fungal elements to grow in an appropriate medium, and lack of sensitivity remains a concern [10] as well as contamination by saprophyte fungi. Molecular approaches such as polymerase chain reaction (PCR) can be used to rapidly identify the presence and strain of fungal elements with high sensitivity. Nevertheless, these approaches are expensive, and local availability varies [11]. Histologic analysis of periodic-acid-Schiff (PAS) stained nail material has demonstrated higher sensitivity for identifying fungal elements compared with KOH and culture [12,13,14,15,16]. However, screening slides for individual fungi to diagnose onychomycosis can be time-consuming for pathologists, and sensitivity remains a concern. There certainly remains room to further improve the diagnosis of fungal nail infections.

Artificial intelligence (AI) and especially machine learning has had a significant impact on the scientific and medical landscape [17,18]. In particular, the subfield of machine learning called deep learning has contributed to significant advances and shown great promise for becoming an integral routine tool in various fields, including medical diagnostics. The idea behind deep learning is to learn complex decisions from the presentation of exemplary data while incorporating minimal human domain knowledge into the learning process. Prior to deep learning, establishing working pattern-recognition algorithms required significant manual feature engineering based on expert knowledge [19].

The application of convolutional neural networks (CNNs), a deep-learning method, for object detection and segmentation has progressed across varied medical domains [20], including radiology, dermatology, and pathology [21,22,23]. Neural networks are composed of layers of processing units called neurons performing mathematical calculations. The parameters (“weights”) of each neuron are adjusted as the network is “trained” on examples of images and their ground truth. Several studies have shown promising results, suggesting that CNNs may aid pathologists in accurately diagnosing skin cancer [24]. A limitation of CNN approaches is the required large amount of data for training, which is not always readily available, especially in the medical field. However, improvement of accuracy and applicability are required before they can be routinely used as a diagnostic assistance tool in clinical routine [25].

U-NET-based segmentation networks (U-NETs) are a subcategory of CNNs specialized in image segmentation. A U-NET returns a segmentation mask where each pixel of the original image has a predicted label, as opposed to the entire image having a single predicted label. U-NETs have been successfully applied in digital histopathology for analyzing whole-slide images (WSIs) of the human tissue [26,27,28,29].

Last year, a study was published examining the potential of a deep-learning algorithm in recognizing tinea in nail clippings of a single institution study, applying a single staining protocol, with four dermatopathologists [30]. We herein present a similar approach applying a U-NET segmentation-based classification approach for the identification of individual fungal elements on the WSI of PAS-stained human nail plates. The slides included came from four different laboratories with different staining protocols and digitized with three different slide scanners. Additionally, the performance of the U-NET was tested in comparison with that of 11 board-certified dermatohistopathologists from different institutions.

## 2. Materials and Methods

### 2.1. Ethics Approval

The study was performed with the approval of the ethics committee of the University of Duisburg-Essen (IRB-number 20–9196-BO).

### 2.2. Image Data Collection

This study used data from multiple centers, and the data were split on a case-by-case basis into a development set for training the machine-learning models and a validation set. The validation set was split into one validation set for evaluating the model and a second validation set for determining the interrater agreement between pathologists and for comparing the model performance with pathologists.

For the development and validation sets for model training and validation, 664 histologic whole-slide images (WSIs) from four different laboratories (MVZ Dermatopathologie Duisburg Essen GmbH, Essen, Germany (*n* = 26); Hautklinik der Universität Duisburg-Essen (*n* = 26); Dermatopathologie bei Mainz, Niederolm, Germany (*n* = 431); and Labor für Pathologie, Salzburg, Österreich (*n* = 181)) were digitized by three different slide scanners (Leica^®^ Aperio AT2, 3dhisttech^®^ Pannoramic 1000, and Hamamatsu^®^ Nanozoomer S360). Slides were selected based on the clinical routine diagnosis. For clinical routine diagnostics, human nail tissue was formalin-fixed, paraffin-embedded, and cut with a thickness of 3 µm to 4 µm. PAS staining was performed on all slides.

### 2.3. Machine-Learning Model and Training

A machine-learning model was developed for predicting the presence of fungus for each pixel in a WSI. The approach consisted of presenting image patches from WSIs to the model and adjusting the parameters according to the guidance from pathologist annotations that outline fungus and other structures (e.g., erythrocytes and artifacts; see Section 2.4 and Table 1).

A U-NET architecture [31] designed for image segmentation with precise localization was used as a basic neural network architecture. Resnet50 [32] represents the backbone of the network, which means that the layer structure of our network matches the convolutional part of the Resnet50 network.

The model was trained for binary segmentation. It classifies each pixel as “Tinea” or “Other” in a prediction map of the same size as the input image. We trained our model using a combination of cross-entropy and dice loss in proportions of 90% and 10%, respectively, and gradually decreased our learning rate in later training epochs using a cosine scheduler. The final model was selected based on the performance on an early stopping set, which was split off the training set. The input size of our model was 224 × 224 pixels, and each patch was augmented by translation, rotation, and color transformations in order to prevent the model from learning spurious correlations. The overall area containing tinea in the dataset and on the slides was very small. To counter class imbalance problems during training, we upsampled patches that contained tinea until 2.5% of all annotated pixels. Similarly, we downsampled the three largest classes (cornified nail, artifact, and air bubble). We did not apply this resampling to the early stopping or test set.

To create a prediction map on the whole slide, we split the WSI into patches. The U-NET creates predictions on each patch of the same size as the input. Finally, all predicted patches are stitched together into a unified prediction map, which can be overlaid on the WSI to aid pathologists.

### 2.4. Annotation Data Collection

A key element for training of AI models is the annotation that guides the learning phase. After digitization, all WSIs were annotated by three board-certified dermatopathologists. Different histologic structures were manually outlined by drawing ground-truth polygons in a proprietary annotation platform. The dermatopathologists were allowed to (sparsely) annotate any region in a slide. After training an initial model, an overlay showing the predictions was made available, and the dermatopathologists were asked to specifically annotate regions where the model’s prediction was incorrect.

The annotations assigned histologic structures to 13 different pathological categories (air bubble, artifact, bacteria, cornified nail, erythrocytes, neutrophils, not pathological, out of focus, parakeratosis, serum, squamous epithelium, tinea (fungal elements), and tissue border). Diagnosis of onychomycosis was confirmed if fungal elements were identified. Fungi could be identified on 36.1% of all WSIs.

We processed the annotations by cropping their image regions into 340 × 340 pixel patches at 0.2743 mpp (micrometers per pixel). Areas outside of the annotation polygons were encoded as “unknown”. This area was excluded from the calculation of the loss of the model, and therefore, the model was allowed to predict anything in the “unknown” area [33]. Overall, the dermatopathologists created 19,278 individual annotation polygons, which were processed into 68,526 patches over 664 WSIs.

We stratified the WSI into three different sets: our patients across training, development, and test sets, while keeping a similar class and laboratory distribution. Our training set consisted of 407 WSIs, resulting in 44,494 patches. Our early stopping development set consisted of 96 WSIs, resulting in 10,926 patches. Our test set consisted of 161 WSIs, resulting in 13,106 patches.

### 2.5. Case-Level Prediction

Pathologists score the presence of tinea per patient, and ideally, a support algorithm will score in the same fashion. For this, we generalized the trained pixel-wise segmentation model to classify individual WSIs by determining if the area segmented as tinea on a slide was larger than a threshold. The threshold set as 0.25% of the tissue on the WSI was predicted as tinea.

This threshold was selected based on evaluating the algorithm’s performance on 78 slides for which an experienced dermatopathologist performed a binary ground-truth scoring. The slides used to select the threshold were an independent set, neither applied for training nor part of the final validation (consensus) set.

### 2.6. Study Evaluation

For the validation of the performance of both the pathologists and algorithm, we selected 74 WSIs with unknown presence of tinea, which were used neither for training nor for identifying the area threshold. The set of 74 slides contained representative shares across the three laboratories and scanners. Eleven board-certified dermatopathologists from 10 different institutions gave a binary decision on the presence or absence of tinea. The selection of “nonclassifiable” was also possible for the dermatopathologists. Screening was carried out using a digital WSI viewer. Similarly, the algorithm scored the 74 slides making binary calls.

Both individual pathologists and the algorithm were compared against the consensus of the 11 pathologists. The consensus was formed by taking the median of the individual pathologist decisions.

## 3. Results

### 3.1. Segmentation Performance

The developed model segmented every tissue pixel in a WSI into tinea (fungal elements) or other tissue applying the described U-NET architecture (Figure 1). This model was evaluated on a holdout validation set where it reached an F1-score of 0.9623. In more detail, the model achieves a high negative predictive value of 99% as well as a high recall of above 95% for negative classes. For the positive class tinea, the predictive value was 85% and the recall was 78.6%. The full pixel-wise performance is shown in Table 2. Examples of correctly classified tinea events are shown in Figure 2, and examples of falsely classified cases, where artifacts or other structures were detected, are shown in Figure 3. Out-of-focus areas frequently impeded correct identification of tinea. Tissue fragmentation and staining artifacts as well as detection of bacteria or serum as tinea were major causes of the algorithm incorrectly making a tinea call (Figure 3).

### 3.2. Case-Level Performance

The segmentation model was generalized to an algorithm that scores cases into positive or negative for tinea (fungal elements). The algorithm’s performance was evaluated against the consensus of 11 board-certified dermatopathologists on a validation set of 74 slides. The algorithm achieved a positive predictive value of 88% and a negative predictive value of 87%. The sensitivity of the model was 94% and the specificity was 77% (Table 3).

### 3.3. Inter-Dermatopathologist Performance

The individual results of the 11 dermatopathologists in accurately detecting fungus on WSIs are depicted in Figure 4 with a median accuracy of 87.84% of all cases. The developed algorithm has an accuracy of 86.49%, demonstrating a comparable performance as the pathologists with four pathologists performing worse and seven pathologists performing on par or better. A ROC curve is shown where the performance of the pathologists is compared with that of the U-NET results (Figure 5). In this representation, the individual calls of not classifiable by dermatopathologists were excluded when calculating the dermatopathologists' performance. We additionally calculated the diagnostic odd ratio (DOR) for each pathologist and for the model (Appendix A).

## 4. Discussion

In this study, we established a deep-learning segmentation-based classification model to detect the presence of tinea on routine PAS-stained slides. After selecting the best model parameters achieved on WSIs for detecting tinea, the deep-learning model demonstrated a comparable consensus performance level to the median of the participating 11 dermatopathologists.

Histologic detection of tinea on PAS-stained slides remains a widely applied approach to detect onychomycosis. Low costs, high specificity, and speed of diagnosis are the advantages [12,14]. If the nail has a high tinea load, diagnosis is simple and quickly made. In cases where there are a few occasions of tinea or no tinea, the identification of tinea becomes very laborious. As “no tinea” is a diagnosis of exclusion, the histopathologist has to carefully assess all the material on the slide at high magnification to make sure that there is not a single hypha present. Even with careful inspection, there is the risk of missing a tinea element and misdiagnosis.

Histologically diagnosing onychomycosis is only reliably possible if PAS staining is performed. PAS staining enhances the contrast to identify fungi, while H&E staining is the routinely performed staining to evaluate other diagnoses such as inflammatory, hereditary, or metabolic disorders and neoplasms [5,6]. Sensitivity for detecting fungi is mainly dependent on the amount of fungi present on the slide, technical preparation, diligence in specimen collection, and practitioners’ experience [14,34,35]. Histologic evaluation can be very time-consuming at an early stage of infection with single or few fungi on the slide. The delay in detecting fungi may enhance costs in repeating analysis with preparation steps and human evaluation [16]. More importantly, missing the identification of fungi at an early stage of disease allows further spread of onychomycosis. An extensive fungal infestation requires systemic antimycotic therapies having a number of potentially serious side effects, including interactions with other medications and severe liver damage. In summary, early reliable detection of nail tinea can substantially reduce costs and patient morbidity.

Tinea detection is a good candidate for an application for which we believe histopathologists will be grateful for the aid from artificial-intelligence-based algorithms. Whereas a diagnosis solely by computer-based algorithms will probably remain problematic (not only from a legal standpoint) [36,37], the proposed model can make a call that the pathologist incorporates into his or her decision. In a positive tinea case, the pathologist can validate the finding and clear the diagnosis. The current process of identifying where the tissue is on the slide and screening for tinea will be eliminated. In particular, for cases with a low number of tinea, this can save the pathologist a considerable amount of time. We believe that in cases where the algorithm does not identify tinea, the pathologist will be required to diligently go through the entire slide to confirm this diagnosis. However, if the computer presents the regions most resembling tinea, and the pathologist identifies tinea, a further detailed assessment of the remaining material is no longer necessary.

We assumed that a simple binary question (tinea or no tinea) would be ideal for establishing a simple yet helpful computer-based diagnostic aid. Unfortunately, a few challenges arose that were not originally expected. Nail material is, by nature, hard material and difficult to process, leading to a relatively high number of artifacts and thicker uneven sectioned slides. Accordingly, almost all digitized scans had areas that were out of focus. Being able to adjust the focus can be instrumental to correctly distinguish tinea and make a correct diagnosis. This was brought up by almost all of the dermatopathologists participating in the study. Applying a “one focus scan” of the slides most certainly impaired the diagnostic capability both for the human pathologists and the deep-learning model. Solutions such as scanning slides more than once at different focus levels might considerably improve the performance and may be worth evaluating.

The results we obtained are promising, especially considering that the case number being well under 1000 is small for training deep-learning machine-learning approaches. The balance between sensitivity (recall) and specificity, as well as precision, depends on the threshold selected for the algorithm. In the current algorithm, we put more emphasis on recall (sensitivity) believing it more important to not miss positive (disease) cases and have the pathologist screen more false positives (lower precision) than the other way around. However, altering the threshold can still be considered. Decreasing the threshold will further increase sensitivity while lowering specificity; on the other hand, increasing the threshold will increase precision and specificity at the cost of sensitivity (recall). As can be expected, our algorithm detected areas as positive in slides classified as not tinea by dermatopathologists. A study assessing if dermatopathologists find reviewing these areas helpful and will on occasion change lead to a change of diagnosis can prove interesting.

Our approach shows a sensitivity value of 93% and a specificity value of 77%. Although specificity is important to accurately confirm the diagnosis of onychomycosis, sensitivity is essential to not miss a fungus and neglect the diagnosis of onychomycosis. In the literature, the sensitivity of identifying fungus with histologic examination has been reported between 80% and 85% [13,14]. The recently published study by Decroos et al. [30] that also assessed tinea detection by AI reported a much higher sensitivity and specificity with an AUC of 0.981. There are some substantial differences in the study design. In particular, all the slides were stained in an automated process with one machine. Additionally, the pathologists in the Decroos et al. study made diagnosis on the analog slides under a microscope, enabling the focus to be readjusted. This eliminated out-of-focus regions commonly observed on our digitized slides, resulting from the focal plane varying due to the thick nail material. Additionally, the pathologists evaluated slides stained in the way they were accustomed to (in their own lab). In our study, the pathologists also were forced to make diagnosis on slides stained with protocols they had not seen before.

We believe that our findings are probably closer to a real-world setting, where deep-learning approaches will need to handle slides stained by different protocols in different labs. On average, the 11 board-certified dermatopathologists in our study achieved a median accuracy of 87.84%, which was comparable with the accuracy of 86.49% of WSIs with fungi detected by the U-NET. We are convinced that the pathologist sensitivity would be higher if they had been allowed to make diagnoses on the analog slides and staining protocols they were accustomed to. The UNET would also likely have performed better if applied to a uniform staining procedure. However, as mentioned, that is not the setting diagnostic algorithms will realistically be faced with.

A limitation of our study is that the deep-learning algorithm was directly compared with diagnosis by pathologists without additional confirmation by other techniques (i.e., PCR or culture). Considering that there is no generally accepted gold standard approach for diagnosing nail fungal infections, comparisons with multiple diagnostic assays might need to be considered in future studies.

Similar to the previous study by Decroos et al. [30], our model had difficulties distinguishing serum and bacteria from tinea. Serum and bacteria were annotated in our training set, though much less frequently as it was less prevalent and not the focus of our study. We are certain that performing additional focused training can greatly improve the algorithms' capability to distinguish tinea from bacteria and serum. With considerably more slides, also events such as nail trauma and hemorrhage (erythrocytes in the nail), bacterial infection (considerable numbers of bacteria), and neoplasms, including potentially the most clinically relevant, acral melanoma, can be picked up by the algorithm. While we are certain that the system can still be greatly improved and expanded to include other diagnoses, we believe that it remains absolutely critical that a human pathologist checks and validates all diagnostic suggestions made.

## 5. Conclusions

Based on the comparable results considering sensitivity in detecting fungus, our established U-NET can already be applied as a supportive diagnostic tool to preselect possible slides with fungi. Those selected slides where a fungus is indicated by the U-NET should be reevaluated by the pathologist to confirm or refute the diagnosis of onychomycosis.

## Figures and Tables

**Figure 1 jof-08-00912-f001:**
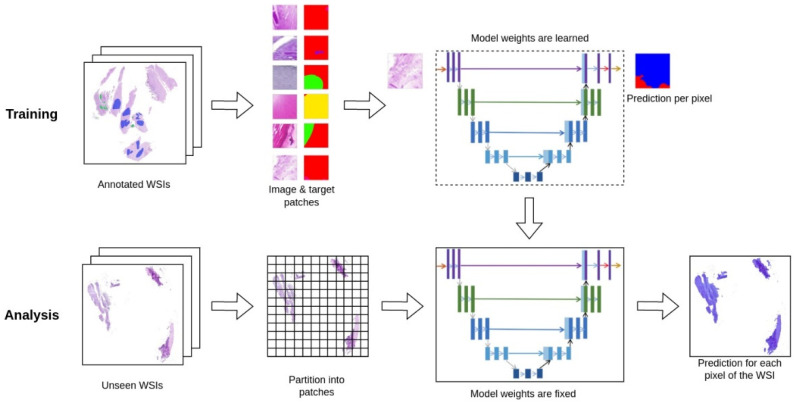
Scheme of the U-NET deep-learning architecture. Demonstrated is the workflow of training our model and using it for analysis. To train our model, we start with WSIs with sparse polygon annotations created by dermatopathologists. The annotations are processed into image and target examples of constant size. These examples are used to train our U-NET segmentation model, which learns to predict either ‘Tinea" or "Not Tinea" for every pixel in the input patch. During this process, the weights of the model are adjusted to improve predictions on training data. To use our model for analysis, we split a WSI into patches and serve each patch to the model to get a prediction. In this process, the model weights are fixed because it is not learning anymore. This also means that the model will give the same prediction for the same input patch. The predicted patches are stitched back together into an image of the same size as the original WSI, so they can be overlaid and used by pathologists to aid them in their diagnosis.

**Figure 2 jof-08-00912-f002:**
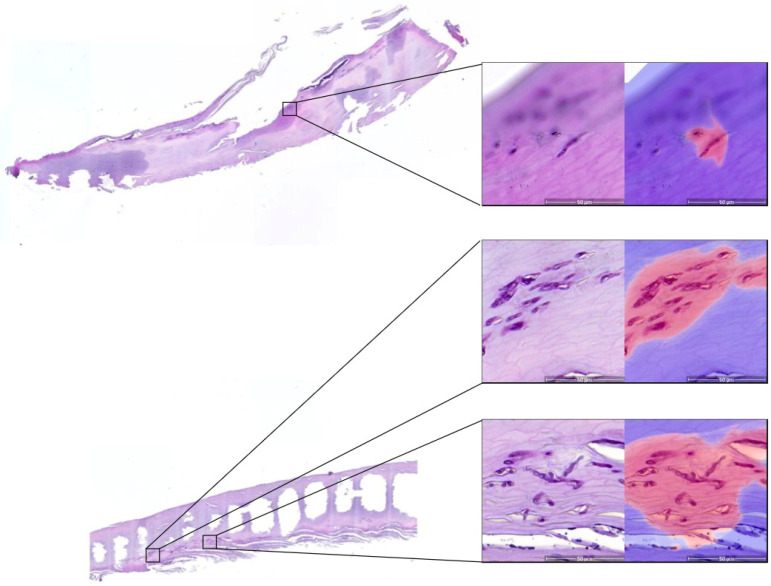
Detection of tinea on whole-slide images. Demonstrated are examples of correctly detected tinea on whole-slide images. Left: an overview of the entire nail section processed; right: zoomed-in examples showing the PAS-stained tinea elements and calls by the algorithm detecting tinea elements, with red pixels representing Tinea. Blurry area on the top image shows areas of the slide where the nail material was not in focus.

**Figure 3 jof-08-00912-f003:**
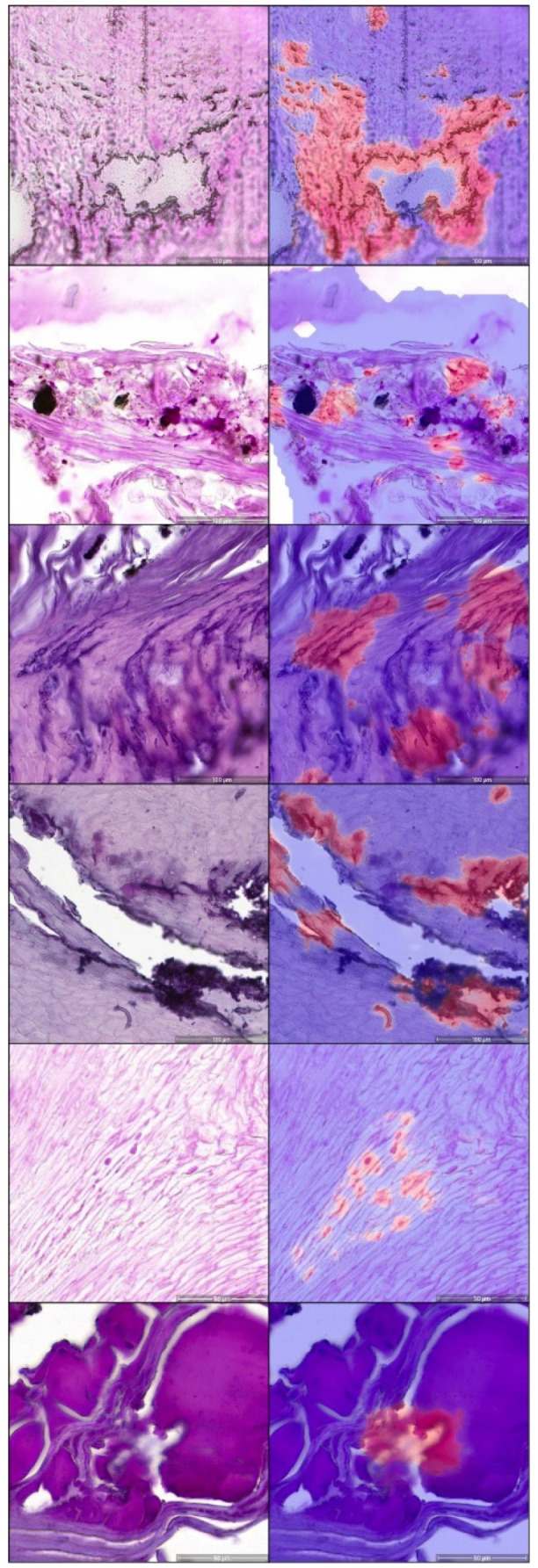
Examples of false-positive tinea calls. Examples where the algorithm made a call of tinea but was not verified by the majority of dermatopathologists are shown. The left column represents PAS-stained slides; the right picture with a heatmap overlay of the U-NET algorithm. The high amount of staining artifacts, common for difficult-to-process nail material, is apparent, which poses a diagnostic difficulty for pathologists and the algorithm alike.

**Figure 4 jof-08-00912-f004:**
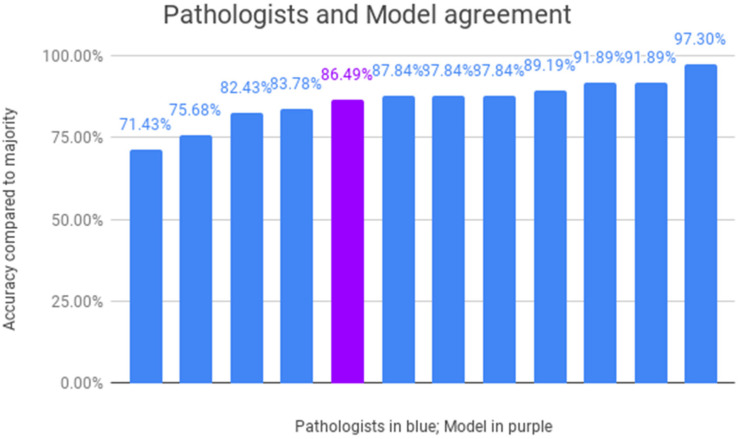
Performance of pathologists and U-NET. Accuracy of each pathologist is depicted in blue. Accuracy of our model is depicted in purple. Correct classification for each case was determined by majority voting across pathologists' representation.

**Figure 5 jof-08-00912-f005:**
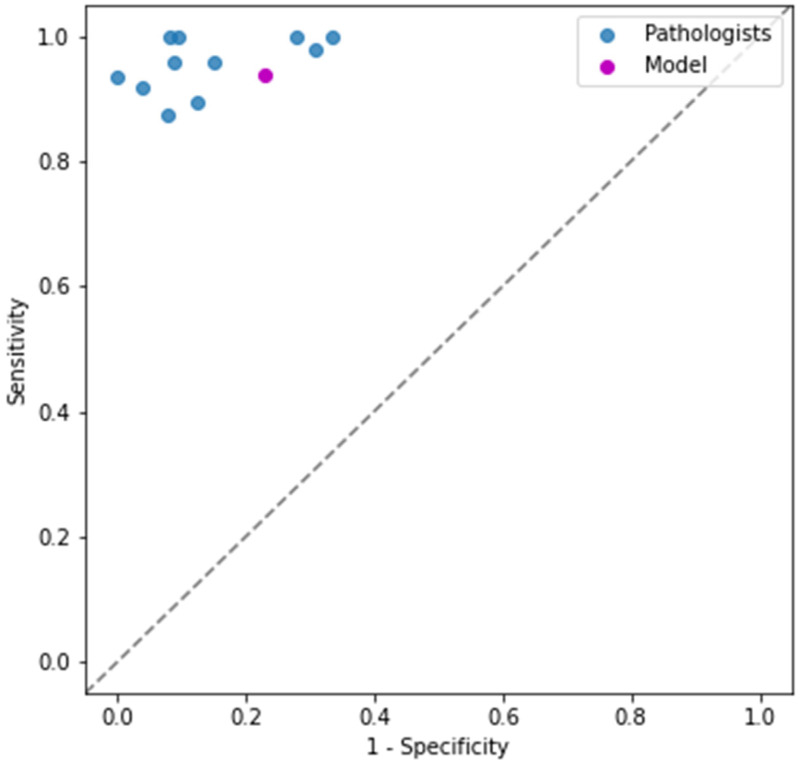
ROC curve comparing 11 dermatopathologists with U-NET algorithm. Demonstrated are individual dermatopathologists as well as U-NET algorithm in terms of sensitivity (true positive rate, recall) and 1-specificity (false positive rate). In the presented plot, cases where pathologists made a "maybe" call were not considered for their performance.

**Table 1 jof-08-00912-t001:** Class distribution of training set after resampling.

Class	Pixel Count (Millions)	Share of Dataset
Cornified nail	166.15	44.27%
Artifact	80.51	21.46%
Air bubble	48.46	12.91%
Out of focus	39.33	10.48%
Serum	15.67	4.18%
Tinea (fungal elements)	9.38	2.50%
Parakeratosis	7.82	2.08%
Tissue border	3.63	0.97%
Squamous epithelium	2.13	0.57%
Erythrocytes	1.02	0.27%
Bacteria	0.99	0.26%
Neutrophiles	0.15	0.04%
Not pathological	0.02	0.01%

**Table 2 jof-08-00912-t002:** Per pixel evaluation. Exact number of predicted pixels can be seen in the two columns. In bold, we highlight correct predictions.

	Tinea	Other	Total	Recall
Air bubble	0	**31,381**	31,381	100%
Erythrocytes	0	**101,001**	101,001	100%
Not pathological	0	**1509**	1509	100%
Out of focus	0	**10,454,027**	10,454,027	100%
Squamous epithelium	0	**38,153**	38,153	100%
Serum	3279	**3,694,931**	3,698,210	99.9%
Cornified nail	224,223	**220,575,318**	220,799,541	99.9%
Artifact	75,452	**23,382,523**	23,457,975	99.7%
Parakeratosis	19,056	**1,315,798**	1,334,854	98.6%
Tissue border	71,738	**4,458,402**	4,530,140	98.4%
Bacteria	38,060	**726,020**	764,080	95.0%
Tinea (fungal elements)	2,456,456	**669,049**	3,125,505	78.6%
Total	**2,888,264**	265,448,112	268,336,376	
Precision	85.0%	99.7%		

**Table 3 jof-08-00912-t003:** Results of the performance per case.

		Positive	Negative	Recall
True label	Positive	45	3	0.94
	Negative	6	20	0.77
	Precision	0.88	0.87	

## Data Availability

The data presented in this study are available on request from the corresponding author.

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
