# Peer review of "Deep Learning Assisted Diagnosis of Onychomycosis on Whole-Slide Images"

_jof, 2022, doi:10.3390/jof8090912_

Round 1
Reviewer 1 Report
My comments are in the attached file.

Author Response
The authors propose a deep learning assisted diagnosis of onychomycosis on whole-2 slide images aiming to help the dermatopathologists as a tool to preselect possible slides with fungi. The article is very a well-writing paper. It is also very interesting. However, in intrinsic value, it does not bring much in the field of medical mycology itself. In below are some remarks which should improve its quality.
We would like to thank the reviewer for her/his kind assessment of our work. We have adapted the manuscript attempting to overcome the mentioned shortcomings.
Point 1: In the “Case-level performance” subtitle on line 253, the authors announced 74 slides but on the table 3 for the validation, only 65 slides are presented. In addition, the values of sensitivity and specificity presented are wrong according to my calculations which found 88.2% sensitivity and 87% specificity..
Authors’ response: We checked the Table and find 74 slides in total, potentially the reviewer did not notice the 6 False-positive and 3 False-negative slides listed in the table. Adding these to the 45 True-positive and 20 True-negative slides, the total of 74 slides is reached. Calculating the sensitivity (True-positive / (True-positive + True-negative)) = 45 / 51 which is 88% (88.23). The specificity (True-negative / (True-negative + False-negative)) = 20 / 23 which is 87% (86.95). So as far as we can see, everything seems to be correct.
Point 2: Abstract
- L32: Please replace “identification” by “diagnose” in the following sentence “Histology remains a frequently applied screening technique to identify onychomycosis”
Authors’ response: We replaced “identify” by “diagnose”. It now reads: “Onychomycosis numbers among the most common fungal infections in humans affecting finger- or toenails. Histology remains a frequently applied screening technique to diagnose onychomycosis.” (ll. 31-32)
- L32-33: Please replace “spores” by “fungal elements” in the following sentence “Screening slides for fungi spores …”
Authors’ response: We replaced “spores” by “fungal elements”. It now reads: “Screening slides for fungal elements can be time consuming for pathologists and sensitivity in cases with low amounts of fungi remains a concern.” (ll.33 -34)
- L65: For its first apparition, please write entirely the word before the abbreviation KOH in the following “ … direct microscopy using KOH …”
Authors’ response: We added the explanation for the abbreviation ”KOH”. It now reads: “Currently, there is no acknowledged standard diagnostic assay, but five methods are widely used: direct microscopy using potassium hydroxide (KOH) staining, fluorescence optical preparation, culture assays, PCR and histologic analysis using PAS stains.” (ll.65 -68)
- L71: Please complete with “as well as contamination by saprophyte fungi” in the following sentence before the full stop “… sensitivity remains a concern(9)”.
Authors’ response: We thank the reviewer for this suggestion and have added it as requested. It now reads: “Cultural growth requires a minimum number of fungal elements to grow in appropriate medium and lack of sensitivity remains a concern (10) as well as contamination by saprophyte fungi.” (ll.65 -68)
L130: Prefer “PAS staining” in the following sentence “PAS-reaction was performed on all slides”
Authors’ response: We replaced “PAS reaction” by “PAS staining”. It now runs: “For clinical routine diagnostics, human nail tissue was formalin fixed, paraffin embedded and sections cut with a thickness of 3µm to 4µm. PAS staining was performed on all slides.” (ll.74-76)
- Table 1: May the authors can put in brackets “fungal elements” after “Tinea”
Authors’ response: We added “fungal elements” after “Tinea” in Table 1.
- L161: Please delete the space in the beginning of the sentence.
Authors’ response: We deleted the space in the beginning of the sentence.
- L179-182: This part in fuzzy. Please reformulate
Authors’ response: Thank you. We restructured the sentence to facilitate understanding its meaning. It now reads: “We stratified the WSI into three different sets: our patients across (training, development, and test sets), while keeping a similar class and laboratory distribution. Our training set consisted of 407 WSIs, resulting in 44494 patches. Our early-stopping, development set consisted of 96 WSIs, resulting in 10926 patches. Our test set consisted of 161 WSIs, resulting in 13106 patches.” (ll.184-188)
Point 3: Results
- Maybe some others diagnostic indices should be calculated as the diagnostic odd ratio and the number need to diagnose among others
Authors’ response: We have performed an analysis of the diagnostic odd ratio and added this figure for interested readers as supplemental figure 1.
Point 4: Discussion
- L365: Please add the reference number just after the authors name as in “Decroos et al. also assessing tinea detection …” and throughout the text.
Authors’ response: We added the reference number after the author name and did the same throughout the whole text. It now reads: “The recently published study by Decroos et al. (30) also assessing tinea detection by AI reports a much higher sensitivity and specificity with an AUC of 0.981.” (ll.373 - 374)
Point 5: Informed Consent Statement
- L415: Please delete the repeated word in “… consent of patients was was …”
Authors’ response: We corrected the sentence. It now reads: “In accordance with the institutional review board statement, informed consent of patients was waived as all WSI were used without information on patient characteristics and were gathered retrospectively and anonymously.” (ll.428 - 430)
Reviewer 2 Report
General comments
The authors use an artificial intelligence (AI), method to facilitate fungal elements in nail plates. This is a promising and interesting study. The major issue in this paper is that the authors consider that all onychomycosis in humans are caused by dermatophyte fungi. This is not true, because between 5-10 % of onychomycosis are due do non-dermatophyte agents, including Candida spp, Aspergillus spp, Fusarium spp, melanized fungi (demaceous), etc. To be more precise the study design should include proved culture cases. The term “tinea”, must be used exclusively for nail dermatophyte infections because it does not englobe the non-dermatophyte onychomycosis agents.
A pilot study with patients presenting culture proved dermatophyte onychomycosis, is desirable.
Please observe that the dermatophytes do not produce “spores” in vivo, but only asexual structures like hyphal fragments. So, try to avoid the term “spores” in all text and change as necessary.
Specific points
P-31. Please check if onychomycosis is the most common fungal infection in humans” According to the literature, 70 to 75% of all women will present one episode of vaginal candidiasis during theirs lives. May be the prevalence of vaginal candidiasis is higher than onychomycosis.
P32 – Please change “screening slides for fungi spores” to screening slides for “fungi elements or fungal structures”
P 301- How “low” is the cost of the AI method compared to the classic histopathologic technique using PAS staining?
Author Response
Response to Reviewer 2 Comments
General comments
The authors use an artificial intelligence (AI), method to facilitate fungal elements in nail plates. This is a promising and interesting study.
Point 1: The major issue in this paper is that the authors consider that all onychomycosis in humans are caused by dermatophyte fungi. This is not true, because between 5-10 % of onychomycosis are due do non-dermatophyte agents, including Candida spp, Aspergillus spp, Fusarium spp, melanized fungi (demaceous), etc. To be more precise the study design should include proved culture cases. The term “tinea”, must be used exclusively for nail dermatophyte infections because it does not englobe the non-dermatophyte onychomycosis agents.
A pilot study with patients presenting culture proved dermatophyte onychomycosis, is desirable.
Authors’ response: We thank the reviewer for his positive feedback. We focused on the most common fungal infections in humans, the dermatophyte. To avoid misunderstandings on the feasibility, we added a statement in the discussion describing that we cannot identify the strain based on the pure appearance of the fungal elements. We additionally agree that a pilot study with patients presenting culture proved dermatophyte onychomycosis would contribute to an improvement of our algorithm. It now reads: “A limitation of our study is that the deep learning algorithm was directly compared to diagnosis by pathologists without additional confirmation by other techniques (i.e. PCR or culture). Considering there is no generally accepted gold standard approach for diagnosing nail fungal infections, comparisons with multiple diagnostic assays might need to be considered in future studies.“ (ll.392-396)
Point 2: Please observe that the dermatophytes do not produce “spores” in vivo, but only asexual structures like hyphal fragments. So, try to avoid the term “spores” in all text and change as necessary.
Authors’ response: Thank you for your explanation. In accordance with the other reviewer´s comments we replaced “spores” by fungal elements.
Specific points
Point 3: P-31. Please check if onychomycosis is the most common fungal infection in humans” According to the literature, 70 to 75% of all women will present one episode of vaginal candidiasis during theirs lives. May be the prevalence of vaginal candidiasis is higher than onychomycosis.
Authors’ response: We agree with the reviewer. Depending on literature, numbers on prevalence vary. Thus we decided to restructure the sentence. It now runs: “Onychomycosis numbers among the most common fungal infections in humans affecting finger- or toenails.” (ll.31-32)
Point 4: P32 – Please change “screening slides for fungi spores” to screening slides for “fungi elements or fungal structures”
Authors’ response: We corrected the terms and avoided to use “spores”. It now runs: “Screening slides for fungal elements can be time consuming for pathologists and sensitivity in cases with low amounts of fungi remains a concern.” (ll.33-34)
Point 5: P 301- How “low” is the cost of the AI method compared to the classic histopathologic technique using PAS staining?
Authors’ response: This is not a question we can currently answer convincingly. The AI detects the PAS stained slide, so the technical tissue preparation costs are exactly the same as when assessed by a human pathologist. The difference would be in terms of pathology personal cost, which varies highly depending on the country. Also there is currently no established pricing for the AI detection. If a stable algorithm is established the technical cost of applying it to new slides would be minute and de facto negligible. However, this does not mean commercial companies would offer the assay at a comparable low price.
Round 2
Reviewer 1 Report
All my suggestions were considered by the authors.
Author Response
Thank you for your feedback. Your comments have significantly improved the quality of our manuscript.
Reviewer 2 Report
Just a one suggestion:
line 77, please change "saprophyte" for "saprobe or saprotic"
Author Response
Thank you for your kind assessment of our work . We have corrected the word in line 77.